# Advancing Our Understanding of Dental Care Pathways of Refugees and Asylum Seekers in Canada: A Qualitative Study

**DOI:** 10.3390/ijerph18168874

**Published:** 2021-08-23

**Authors:** Nazik M. Nurelhuda, Mark T. Keboa, Herenia P. Lawrence, Belinda Nicolau, Mary Ellen Macdonald

**Affiliations:** 1Faculty of Dentistry, University of Toronto, 124 Edward St, Toronto, ON M5G 1G6, Canada; nazik.suleiman@dentistry.utoronto.ca (N.M.N.); herenia.lawrence@dentistry.utoronto.ca (H.P.L.); 2Faculty of Dentistry, 500-2001 McGill College, McGill University, Montréal, QC H3A 1G1, Canada; mark.keboa@mail.mcgill.ca (M.T.K.); belinda.nicolau@mcgill.ca (B.N.)

**Keywords:** oral health, refugees, Canada, migrants, dental care

## Abstract

The burden of oral diseases and need for dental care are high among refugees and asylum seekers (humanitarian migrants). Canada’s Interim Federal Health Program (IFHP) provides humanitarian migrants with limited dental services; however, this program has seen several fluctuations over the past decade. An earlier study on the experiences of humanitarian migrants in Quebec, Canada, developed the *dental care pathways of humanitarian migrants* model, which describes the care-seeking processes that humanitarian migrants follow; further, this study documented shortfalls in IFHP coverage. The current qualitative study tests the pathway model in another Canadian province. We purposefully recruited 27 humanitarian migrants from 13 countries in four global regions, between April and December 2019, in two Ontario cities (Toronto and Ottawa). Four focus group discussions were facilitated in English, Arabic, Spanish, and Dari. Analysis revealed barriers to care similar to the Quebec study: Waiting time, financial, and language barriers. Further, participants were unsatisfied with the IFHP’s benefits package. Our data produced two new pathways for the model: transnational dental care and self-medication. In conclusion, the dental care needs of humanitarian migrants are not currently being met in Canada, forcing participants to resort to alternative pathways outside the conventional dental care system.

## 1. Introduction

By mid-2020, the number of displaced people globally had exceeded 80 million [1]. Among this number are nearly 26.3 million refugees, “individuals who are unable or unwilling to return to their country of origin owing to a well-founded fear of being persecuted for reasons of race, religion, nationality, membership of a particular social group, or political opinion,” and 4.2 million asylum seekers, individuals who are seeking refugee status [2]. Canada has welcomed 1,088,015 humanitarian migrants (refugees and asylum seekers) since 1980 [3].

Globally, the burden of oral diseases is high in humanitarian migrants, followed by a high need for dental care [4]. In Canada, one study reported 85% of adult Bhutanese refugees, who originated from refugee camps in Nepal, had one or more teeth with untreated decay, compared to 20% of the Canadian population [5]. Refugees, in general, are less likely to access oral healthcare services, and their first dental visit will often be due to tooth pain [6]. The inverse care law in dental care, which describes the principle that those who need treatment the most have the least access to care, continues to exist in Canada [7].

Universal oral health coverage will be achieved when all individuals have access to needed oral health services without financial hardship [8]. Dental care services in Canada, unlike medical care, are not publicly funded; instead, they are primarily consigned to individual responsibility. Canada is among the highest countries in mean per capita spending on dental care, yet among the lowest in terms of public share [9]. Dental expenses are often covered through private insurance; almost two-thirds of Canadians (64.6%) have private dental insurance [10], otherwise citizens rely on out-of-pocket payments, and/or government assistance programs (e.g., for children in some jurisdictions: Healthy Smiles Ontario; low-income seniors: Ontario Seniors Dental Care Program) [11].

The Migrant Oral Health Project (MOHP) is a Canadian research team concerned with the oral health of humanitarian migrants. In 2019, Keboa and other MOHP members used focused ethnography to investigate how humanitarian migrants experienced and accessed dental care in Montreal, Quebec [12]. Informed by the public health model of the dental care process, [13] the team recruited 25 humanitarian migrants from ten countries representing Latin America, North Africa and the Middle East, Eastern Europe, Southeast Asia, and Sub-Saharan Africa. With data from interviews and observations, the authors developed the *dental care pathways of humanitarian migrants* model, showing four pathways: A conventional pathway to the dental clinic, a pathway to the clinic through non-dental health professionals, a pathway to the dental clinic through non-healthcare providers, and seeking advice and medication from abroad instead of seeking help in a health clinic in Canada.

In that previous study, the participants were covered for basic dental care by the federally-funded Interim Federal Health Program (IFHP). This program, which has undergone several modifications in the past decades, provides humanitarian migrants with basic health services, including dental, until they become eligible for provincial or territorial health insurance. One goal of Keboa et al.’s study was to understand what participants felt about the dental coverage provided by the IFHP, which had been severely reduced in 2012 and 2014 [14]. In 2016, IFHP was reformed to cover more services—but which required preauthorization - such as simple fillings and dentures. Root canal treatments, sedation, orthodontic treatment, scaling, and root planning were not covered. In 2020, more treatment options were added to the preauthorization requirement list, but with a maximum dollar amount limit [15].

The aim of our current study, which is part of a larger mixed-methods research program [16], was two-fold: To test the pathway model by exploring pathways to oral health care among humanitarian migrants in another Canadian province (Ontario); and to explore the perception of humanitarian migrants of the IFHP which had been reformed in 2016 to provide more extensive—yet still basic—dental services to its beneficiaries.

## 2. Materials and Methods

### 2.1. Study Participants and Recruitment

We purposefully recruited humanitarian migrants from global regions that corresponded to Keboa et al.’s sample: Middle East, Central and South America, South Asia, and Africa. Participants were recruited from three community organizations providing services to humanitarian migrants: one in Ottawa, and two in Toronto. We used brief information sessions with potential participants and word-of-mouth to recruit participants. Using purposeful sampling, we invited participants who had arrived in Canada as refugees or asylum seekers (referred to as ‘refugee claimants’ in Canada), were 18 years old and above, and had previous experience with the dental care system in Canada. We purposefully sought a sample varied in age, gender, and country of origin. Participants were compensated for travel expenses and their time.

### 2.2. Methods for Data Generation

We developed our initial semi-structured interview guide from the main findings of Keboa et al.’s study. Focus group discussions were preceded by a short questionnaire gathering participants’ sociodemographic information. Focus group topics included: Oral health knowledge, hygiene habits, and past experiences with the dental care system in Canada. After an initial conversation, we explicitly brought in the dental care pathway model to the discussion, to encourage participants to reflect on their personal experiences and challenge, and to enhance the model. We adapted our questions as we proceeded with iterative analysis during data generation.

We held four focus groups, one each in English, Spanish, Arabic, and Dari. Data were generated between April 2019 to December 2020. We chose focus groups as our method to provide participants with an opportunity to exchange views and discuss ideas, and help us understand how humanitarian migrants seek care for oral health issues. Team members facilitated the English and Arabic focus group discussions (M.T.K. and N.M.N., respectively) and trained a Spanish-speaking and a Dari-speaking facilitator for the remainder. All facilitators were migrants to Canada, and one was a refugee. The focus group discussions lasted between 45 and 90 min. All discussions were held in private rooms in each of the three community organizations. Fortunately, while data generation was interrupted by the COVID-19 pandemic, we reached data sufficiency prior to lockdown.

### 2.3. Data Management and Analysis

The focus groups were audio-recorded, and recordings were stored on a secure server. Recordings were transcribed verbatim and translated to English for analysis. Our study included multiple investigators from different backgrounds to foster reflexivity and dialogue. We began our analysis by building a deductive coding structure based upon the interview guide and the dental care pathway model. We then proceeded with both deductive and inductive analysis, with one team member going line-by-line to build codes that enhanced or challenged the coding structure. After completing a comprehensive review of all data, codes were merged into themes and shared with the broader team for discussion and confirmation. This article adheres to the Standards for Reporting Qualitative Research (SRQR) guidelines [17].

### 2.4. Ethical Approval

The study was approved by the University of Toronto’s Research Ethics Board (RIS Protocol No. 36911). Written informed consent was obtained from all participants. The facilitator explained the participants’ rights, the procedures to be undertaken, purpose of the study, expected duration of study, potential risks and benefits of participation and confidentiality of personal identification, in the participants’ language of preference.

## 3. Results

We begin the results by describing the participants’ profiles, and their experiences with facilitators and barriers to dental care access. Following, we describe how the data contributes two additional dental care pathways and outcomes of care to Keboa et al.’s model.

### 3.1. Participant Profile

We held four focus group discussions with 27 participants from thirteen countries: Palestine, Syria, Iraq, Mexico, El Salvador, Honduras, Venezuela, Afghanistan, Iran, Eritrea, Somalia, Nigeria, and the Democratic Republic of the Congo. Participants’ age ranged from 28 to 62 years. Most participants were women (65%), unemployed (78%), completed high school or more (80%), and had an annual income that is below minimum wage (78%). Table 1 describes the sociodemographic profile of the study participants.

### 3.2. Facilitators to Care

As described in Figure 1, the starting point for the participants’ dental care pathway was ‘Humanitarian migrants in need for dental care’. Our results suggest that participants were aware of when they needed to seek professional oral health care. Two relevant domains, identified as facilitators to care, were generated by the data analysis, as follows:

#### 3.2.1. Importance of Oral Health

Many of the participants spoke of oral health as a fundamental component of overall health. As one participant said: *“The centre of all body functions is the mouth. So, the hygiene for oral cavity has to be kept in mind.”* She then recited a poem by the Persian poet Saadi: *“If one organ is in pain, other organs are not at peace as well.” (48-year-old Iranian woman).*

Participants showed a solid understanding of the importance of oral health. The phrase ‘good oral health’ was described with oral hygiene habits, such as brushing teeth regularly, visiting a dentist for dental check-ups, and using dental floss; ‘poor oral health’ was commonly described as bad breath, bleeding gums, missing and crowded teeth.

For children, mothers reported that ‘good oral health’ meant brushing teeth after breastfeeding. Overall, child dental care was not a concern for participants: “*They get the best treatment here.” (41-year-old Mexican woman).*

#### 3.2.2. Changes in Oral Hygiene Practice

Participants’ felt that their knowledge of oral hygiene habits did not change after arriving in Canada; however, they reflected that their practices did. They said they are now more careful about following good hygiene measures in Canada because of an anticipated difficulty in dental care access. A man in his 40s from Venezuela said: *“I think we are more alert because it is very expensive [to treat dental problems] so we take all the precautions.”* He added: “*In our countries, we don’t give them too much importance but once we are here, we want to prevent [dental disease] with cleaning and dental tools because we don’t want anything bad to happen. And if it does, we have to have [have] at least $1000 saved just in case.”*

Overall, participants’ understanding of the importance of oral health, their overall knowledge of oral health and hygiene, and their awareness of the local dental care system made them likely to identify their need for dental care.

### 3.3. Barriers to Care

#### 3.3.1. Financial Constraints

Finances were considered a major barrier by all participants, and cost was commonly the first issue raised. In particular, participants were unsatisfied with the IFHP benefit package available to them. As a young Syrian woman said, *“General health coverage [by IFHP] is good, but for dentistry it is lacking. This being a country for refugees, there is a big shortage.”* A Somali woman reflected that IFHP is not generally accepted: *“I have [IFHP insurance] for one year. Then they say this one [IFHP] cannot cover you.”* A Nigerian woman was also concerned about the coverage: *“To remove, or to shape your teeth, or to polish, is expensive.”*

A middle-aged Afghani woman summed up the issues: *“I had insurance [IFHP] at that time, I consulted the dentist. My friends referred me to a dentist in [a Toronto] area. But it was really very expensive. As I mentioned before, only one root canal treatment [not covered by IFHP] cost 1200 dollars.”*

Financial barriers also affected the frequency of dental visits by participants. Regular visits to dentists for preventive reasons were reported as common practice in their home countries, but in Canada, the behavior has become infrequent for some, and absent for most: *“We used to visit a dentist every 3 to 4 months, but when we decided to migrate it turned out to be very expensive [in Canada]. Therefore, from going every 4 months now we will only go once a year.” (41-year-old Venezuelan man).*

#### 3.3.2. Waiting Time for Receipt of Dental Care

Participants complained about the long time they had to wait before they could schedule an appointment. They followed the conventional pathway in the model; yet, theydid not always receive care within a window of time they felt was reasonable. As a woman from Afghanistan said, *“If I tell the dentist that it is an emergency, I will still have to wait for at least 20 days for my appointment.”*

Others reported that long waiting time was a barrier because they found it difficult to keep track of their appointments. As newcomers, they were overwhelmed by the list of appointments to which they had to attend. As a result, they were not able to obtain the treatment they needed: *“The first time I went, they gave me a 2 months appointment and eventually I forgot I had that appointment.” (30-year-old Nigerian woman).*

#### 3.3.3. Language Barriers

Language barriers were found to contribute to a reduced satisfaction with services provided. A 41-year-old Venezuelan man said: *“A barrier is the language and technical words that we don’t know, and which make communication difficult.”* Others have also echoed a similar concern.


*“The first problem I had was with the receptionist because she said I needed a translator even though I told her I understood. Another woman helped me, she was kinder, filled out my forms and gave me an appointment.” (37-year-old Mexican man).*



*“The first thing I had to do is to fill out all the paperwork and I felt like it was overwhelming.” (41–year-old Venezuelan woman).*


Some participants received information on navigating the system from the newcomer organizations with which they were registered. Their experiences of this support varied, however; some were satisfied, while others felt that they did not receive useful information. A young Honduran man said that: *“We had to ask someone else outside [the organization] because nobody helped us there.”* A Mexican participant highlighted the same concern, saying: *“For example, we asked our case worker about where to go for a dental appointment and she said that we had to find out on our own.”*

### 3.4. Two New Pathways

Together, these barriers culminated in the creation of two new pathways in the dental pathway model, as well as new outcomes to the care received (Figure 1).

#### 3.4.1. Transnational Dental Care

The first new pathway is *transnational dental care*. Given the prohibitive cost of treatment, combined with trust in familiar dental services, participants compared local dental care unfavorably to the care they received in their home countries:


*“In my country, we have better access and [treatment] is cheaper.” (56-year-old Venezuelan woman).*



*“In my country there isn’t much protocol; here you have to go through many steps.” (31-year-old Venezuelan man).*



*“In Egypt and Syria, dentists are very clever.... the filling does not come out.” (27-year-old Syrian woman).*



*“I had a crown done for me back home in Iraq before 2003. Until today, it is the same colour, the same colour as my teeth.” (47-year-old Iraqi woman).*


Thus, participants said they were prepared to travel outside Canada, when immigration laws allowed, to receive dental treatment. This sentiment was summed up by a middle-aged Iraqi woman: *“There are some people that I know who said they would buy an air ticket purposely to fix their teeth. They had a big problem with their teeth. I heard them, they said we pay for an air ticket, and we get to see our family also, and still, it will be cheaper than treating our teeth in Canada.”*

#### 3.4.2. Self-Medication

The second new pathway is *self-medication*. Some participants sought to treat themselves, often as a temporary remedy to their pain, until they could afford professional care. A young Afghani woman spoke of using “*garlic to relieve the pain”* while waiting for care; a Mexican man spoke of *“pain killers”* being cheaper, and thus more accessible, than dental care.

### 3.5. Outcomes of Care Received

The prohibitive cost of dental treatment was found to influence the decisions made by the participants. Some participants opted to decline the recommended treatment, and this is referred to as ‘No Care’ in the original model. One participant explained: *“I was told I would have to pay $1500 for the extractions. I have other expenses to do and that is why I suspended that for the moment, but to come to this point it took me almost the year that I’m here.”*

Another participant, after receiving help with navigating the appointment system, was not able to receive the needed treatment: *“I have my own translator, but even with that it has been really hard. He made all the appointments, he called, he asked everywhere, and we haven’t found a solution yet because I have a jaw disorder, so they told me that I have to pay $3000 and that is where we stopped. I have insomnia and headaches.” (37-year-old Honduran woman).*

An addition to the pathway model (Figure 1) from our data pertains to outcomes of care received after meeting with the dental care provider. *‘Complete care’* is when a participant received the whole treatment as recommended by the dental care provider. The second and new outcome was to opt for *‘partial treatment’*: Here, the care was received, but was incomplete. This outcome is summarized by a 35-year-old Salvadorian man as follows:


*“They did two fillings but one of then was very deep. When they did it, I think they touched the nerve… that in a week I couldn’t bear the pain. Three weeks passed, I went to see the dentist, he told me it was sensitivity and that I had to go somewhere else to get it fixed because he couldn’t. He gave me a card, so he referred me to another doctor where I had to pay. I had the pain for a month, I had to pay for a root canal treatment $1400 and I haven’t come back for the crown yet because it is another $900.”*


Similarly, an older Palestinian woman reported that a dentist *“re-glued the front denture.”: “When I arrived in Toronto my daughter was looking for a job to make ends meet. She found a job with a dentist. She told him about my situation so he told her that she can bring me to the clinic. He was kind and fixed my teeth at no cost. He said that this was a temporary solution and that in about a year they might come off again.”*

## 4. Discussion

Our focus groups suggest that humanitarian migrants arrive in Canada with an awareness of when they needed to seek professional dental care; this awareness facilitated their care seeking. However, once seeking care, they were met with waiting time, financial, and language barriers. As a result, some study participants opted to receive only partial care, some self-medicated, and many considered transnational treatment as a viable option to accessing care in the future.

These results confirm, as well as extend, Keboa et al.’s (2016) earlier work in important ways. To begin, the participants’ oral health knowledge and practices confirm that the personal identification of the need for treatment serves as a valid starting point for the dental care model. Further, while the IFHP program had been amended in 2016, providing more benefits than for participants in the Montreal study, participants were still unsatisfied with the coverage. A systematic review and meta-analysis found that covering individuals with dental insurance has a significant effect on increasing utilization in dental care [18]. Further, evidence of financial barriers experienced by the general population, including migrants, have been demonstrated elsewhere [19,20]. The Ontario government launched the Healthy Smiles Ontario program for children to bridge the gap for low-income families, including humanitarian migrants [21]. For adults, the federal government is progressively improving the IFHP [14]; however, as our study shows, these improvements have not yet met the needs of humanitarian migrants. These financial barriers were the commonly reported reason behind opting out of, or receiving partial, care.

Screening migrating populations when they enter their host country is a powerful tool to promote oral health. Refugee populations are disproportionately affected by common oral diseases [4], and upon arrival in the host country, they often need professional dental attention. The process of identifying treatment needs can be proactive and commissioned through government-funded oral health screening programs with stable referral systems in place [22]. Such programs are not common practice; only four countries, namely, Armenia, Canada, Iran, and Italy, report obligatory oral health screening upon arrival [22]. While Canada does provide screening, it does not have a referral system in place for the screened individuals [23]. Therefore, newcomers must rely on their own judgment and navigation capacity to seek professional care. Findings from our study suggest that since participants understand the difference between good and bad oral health, and are aware of oral hygiene habits that promote oral health, they can actively seek dental care when they are in need. This passive approach to seeking dental care, wherein an individual is expected to navigate the dental care system upon their own perception of the need for treatment should be replaced by a more efficient public health care system that actively identifies and directs people to dental care providers when they need care.

Transnational dental care was not reported in the Montreal study. Our participants reported their interest in traveling outside Canada to get affordable treatment. This strategy has been described as “immigrants seeking dental care across national borders in the form of dental tourism or while travelling(*sic*) to their country of origin” [24]. Calvasina et al. analyzed *the Longitudinal Survey of Immigrants to Canada 2001–2005* and found that immigrants lacking dental insurance and those reporting dental problems were more likely to report transnational dental care utilization [24]. Another study, exploring dental care pathways in a cohort of Chinese immigrants in Montreal, also found that consulting a dentist during a return visit to China was common [25]. Importantly, it is not only immigrants who seek dental tourism; one study examined low-income Canadians seeking less expensive care in Mexico [26]. No study on dental tourism to date has specifically included humanitarian migrants.

This evidence around transnational dental care demonstrates weaknesses in the Canadian dental care system in providing publicly-funded dental care to vulnerabilized groups [27]. The situation for humanitarian migrants is complicated, as their movement across borders is restricted. For them to pursue transnational treatment, they would need to complete official travel documents, which could mean a delay of treatment, even for years. Dental diseases are mostly progressive, and if left untreated, can result in hospitalization. Therefore, this option of transnational dental care utilization, although considered by our participants, could worsen their oral health situation.

The original model included the pathway, ‘advice and medication from abroad’ (Keboa et al. 2019). Our participants’ self-medication behavior was not necessarily informed from abroad, however. Self-medication, which has been defined in the literature as “the taking of drugs, herbs or home remedies on one’s own initiative, or on the advice of another person, without consulting a doctor” [28] is, therefore, an important addition to the model. It is common for people to self-medicate for dental problems. For example, a Nigerian study found the prevalence of self-medication for dental problems high, with using both orthodox and unidentified traditional drugs [29]. Similarly, in India, the practice of self-medication, using an array of drugs (e.g., analgesics, native herbs, antibiotics), was found to be motivated by friends and relatives [30]. In a Cameroon study, the majority of respondents self-medicated for oral health problems, guided by advice from relatives [31].

Self-medication can be a dangerous practice because using some medicines without medical guidance may result in inappropriate, incorrect, or undue therapy, missed diagnosis, delays in treatment, pathogen resistance and increased morbidity [32]. However, it can be seen as the “desire and ability of people/patients to play an intelligent, independent, and informed role, not merely in terms of decision-making but also in the management of those preventive, diagnostic and therapeutic activities which concern them [32].” Improving the knowledge of humanitarian migrants about the possible harms of self-medication is important to both limit microbial resistance of antibiotics, as well as to ensure personal safety.

Although our study participants described waiting time and language as barriers seemingly disconnected from financial barriers, we believe that these experiences were connected. Participants had to wait because only a limited number of dental clinics provided services that were covered by IFHP. In addition, since financial barriers to dental care access are shared by Canadians at large, and in particular, low-income individuals, we understand that humanitarian migrants’ experiences may be shared by other Canadians. However, humanitarian migrants are forced to leave their country to escape war, persecution, or natural disaster, and therefore are more vulnerable to developing severe dental problems. Thus, they need timely access to dental care in their new home.

As with any study, our study has important strengths and limitations. To begin, our work builds on the findings of a previous study, providing further insight into the dental care experiences of humanitarian migrants in Canada. To the best of our knowledge, this is the first study to report on humanitarian migrants from Ontario. While we could not include participants from all countries of origin of humanitarian migrants arriving in Canada, we interviewed humanitarian migrants from the four main global regions of origin, and our focus groups were facilitated in the mother tongue of participants. The diversity in our study population provides much detailed information to understand complex issues. We acknowledge the limitation of qualitative study designs; as it is not possible to generalize the findings to the study population, our goal was to describe the experience of humanitarian immigrants. The use of focus group discussions was a productive choice as focus groups allow participants to jointly brainstorm and debate ideas, opinions, and recommendations.

## 5. Conclusions

The dental needs of humanitarian migrants in Ontario are not being met by local dental services, and our participants were not satisfied with IFHP coverage. Although they had good oral health knowledge, access to dental care for our participants was hindered by financial barriers, waiting time, and challenges navigating the oral health system. Adding to the original pathway model, participants in our study opted to receive partial care, and endorsed self-medication and transnational dental care as additional dental care pathways.

To date, our MOHP team has canvassed the perspective of humanitarian migrants from central Canada (Quebec and Ontario), the locations of the majority of humanitarian migrants. Future work should explore experiences of humanitarian migrants in Western and Eastern regions, and test the model in other countries with similar dental care systems.

## Figures and Tables

**Figure 1 ijerph-18-08874-f001:**
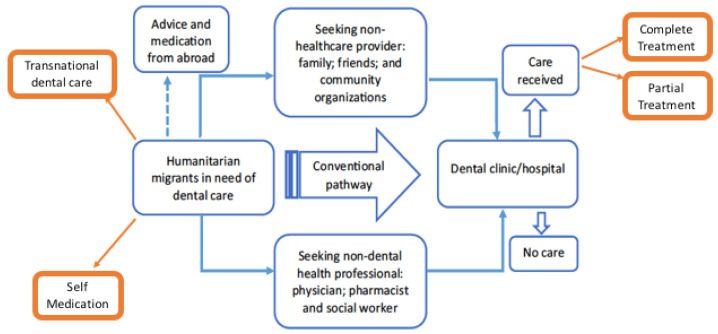
An updated dental care pathways model of humanitarian migrants. Modifications based on the present study are highlighted in orange [12].

**Table 1 ijerph-18-08874-t001:** Sociodemographic profile of study participants.

Characteristics	Middle East (Arabic)	Central and South America (Spanish)	South Asia (Dari)	Africa (English)	Total
Country of origin	Palestine, Syria, and Iraq.	Mexico, El Salvador, Honduras, and Venezuela.	Iran and Afghanistan.	Eritrea, Somalia, Nigeria, and the Democratic Republic of the Congo.	13
Number of participants	7	9	4	7	27
Age Range (years)	28–62	31–56	29–51	21–49	28–62
Gender					
Man	1	5	1	2	9
Woman	6	4	3	5	18
Completed Education					
Less than high school	3	0	1	4	8
High school and above	4	9	3	3	19
Immigration status upon arrival in Canada					
Refugee	3	1	4	4	12
Refugee Claimant (Asylum seeker)	4	8	0	3	15
Current immigration status					
Refugee	6	1	4	6	17
Refugee claimant (Asylum seeker)	1	8	0	1	10
Employment status					
Full or part time	1	3	2	0	6
Unemployed	6	6	2	7	21
Income					
≤$20,000	6	6	2	7	21
>$20,000	1	3	2	0	6
Spent time in a refugee camp					
Yes	3	2	0	0	5

## Data Availability

The data presented in this study are available on request from the corresponding author. The data are not publicly available as this could compromise the privacy of research participants.

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
