# Peer review of "Advancing Our Understanding of Dental Care Pathways of Refugees and Asylum Seekers in Canada: A Qualitative Study"

_ijerph, 2021, doi:10.3390/ijerph18168874_

Round 1
Reviewer 1 Report
The authors wrote about a very interesting but often neglected topic connected with an oral medical health ina group pf refugees and aaylum seekers in Canada.
In my opinion, the results and conclusins should be shared to the departments that organize health system in Canada. I suggest to accept this paperAuthor Response
We are pleased that Reviewer 1 enjoyed our manuscript. We intend to share our study findings via several channels, such as scientific publications in peer reviewed journals, presentations in scientific conferences and policy briefs. At the end of the whole project, we are planning to hold a knowledge translation workshop with knowledge users who are likely to make use of the research results to inform their decisions and practices (e.g., Office of the Chief Dental Officer of Canada).
Reviewer 2 Report
The paper shows that the data expand the findings in the previous study and two pathways are added among small number of participants in a qualitative study.
I understand the qualitative study. However, there are some issues.
Then, the paper should be revised.
- Please follow the SRQR checklist and add the list in the paper. For examples, title [Concise description of the nature and topic of the study Identifying the study as qualitative or indicating the approach (e.g., ethnography, grounded theory) or data collection methods (e.g., interview, focus group) is recommended], Researcher characteristics and reflexivity [Researchers' characteristics that may influence the research, including personal attributes, qualifications / experience, relationship with participants, assumptions and / or presuppositions; potential or actual interaction between researchers' characteristics and the research questions, approach, methods, results and / or transferability], sampling strategy [How and why research participants, documents, or events were selected; criteria for deciding when no further sampling was necessary (e.g., sampling saturation); rationale], the specific number of approval by an appropriate ethics review board, Data collection methods [Types of data collected; details of data collection procedures including (as appropriate) start and stop dates of data collection and analysis, iterative process, triangulation of sources / methods, and modification of procedures in response to evolving study findings; rationale], Data collection instruments and technologies [Description of instruments (e.g. interview guides, questionnaires) and devices (e.g. audio recorders) used for data collection; if / how the instruments(s) changed over the course of the study], and others should be considered.
- Please clarify the main findings and evidence (e.g. quotes, field notes, text excerpts, photographs) to substantiate analytic findings using a figure or table. The authors add the orange area in the Figure 1. However, it does NOT link to the summarized data of interviews (the participants are not satisfied with IFHP). The readers can’t find the data; how many participants are not satisfied, how many participants received partial care and how many participants do self-medication. Please add more details in the result section.
- Why don’t the authors add the data of previous study and analyze them in a cross-sectional study?
Author Response
We thank Reviewer 2 for reviewing our manuscript.
The paper shows that the data expand the findings in the previous study and two pathways are added among small number of participants in a qualitative study. I understand the qualitative study. However, there are some issues. Then, the paper should be revised.
- Please follow the SRQR checklist and add the list in the paper.
We revised the manuscript to ensure that all the SRQR checklist items are clearly addressed in the paper. Also, the following statement has been added to section Page 5 line1:
This article adheres to the Standards for Reporting Qualitative Research (SRQR) guidelines.
- For examples, title [Concise description of the nature and topic of the study Identifying the study as qualitative or indicating the approach (e.g., ethnography, grounded theory) or data collection methods (e.g., interview, focus group) is recommended],
Our new title has been improved to read: Advancing our understanding of dental care pathways of refugees and asylum seekers in Canada: a qualitative study.
- Researcher characteristics and reflexivity [Researchers' characteristics that may influence the research, including personal attributes, qualifications / experience, relationship with participants, assumptions and / or presuppositions; potential or actual interaction between researchers' characteristics and the research questions, approach, methods, results and / or transferability],
The following statement has been added to the ‘Data management and analysis’ section on Page 4 Line 27: “Our study included multiple investigators from different backgrounds (Dentistry, Public health, Epidemiology, and Anthropology) which fostered reflexivity and dialogue. This diversity in values, perspectives and assumptions fostered both complementary and divergent understandings of the study results which were resolved through consensus.
- sampling strategy [How and why research participants, documents, or events were selected; criteria for deciding when no further sampling was necessary (e.g., sampling saturation); rationale],
We built our sample to correspond to Keboa et al’s sample in the Montreal Study since we worked with their pathway model. Our sampling strategy is outlined in the “study participants and recruitment” section on page 3 line 26 to page 4 line 3, as follows:
We purposefully recruited humanitarian migrants from global regions that corresponded with Keboa et al’s sample: Middle East, Central and South America, South Asia and Africa. Participants were recruited from three community organizations providing services to humanitarian migrants, one in Ottawa, and two in Toronto. We used brief information sessions with potential participants and word-of-mouth to recruit participants. Using purposeful sampling, we invited participants who had arrived in Canada as refugees or asylum seekers (referred to as ‘refugee claimants’ in Canada), were 18 years old and above, and had previous experience with the dental care system in Canada. We purposefully sought a sample varied in age, gender and country of origin.
- the specific number of approval by an appropriate ethics review board,
For this study, we obtained two ethics approvals: one from University of Toronto and another from McGill University. This is highlighted on page 5 line 6.
- Data collection methods [Types of data collected; details of data collection procedures including (as appropriate)
This following statement in the ‘data generation’ section on Page 4 line 6 - 12 describes the methods and interview topics:
We developed our initial semi-structured interview guide from the main findings of Keboa et al’s study. Focus group discussions were preceded by a short questionnaire on participant socio-demographic information. Focus group topics included: oral health knowledge, hygiene habits, and past experiences with the dental care system in Canada. After initial conversation, we explicitly brought in the dental care pathway model to the discussion to encourage participants to reflect on their personal experiences, and challenge, and to enhance the model. We adapted our questions as we proceeded with iterative analysis during data generation.
- start and stop dates of data collection and analysis, iterative process, triangulation of sources / methods, and modification of procedures in response to evolving study findings; rationale], if / how the instruments(s) changed over the course of the study], and others should be considered.
Data was generated between April 2019 and December 2020 (Page 4 Line 14). We adapted our questions as we proceeded with iterative analysis during data generation (Page 4 Line 12). Initially, we were asking participants to comment directly on findings from the Montreal Study. As we proceeded with data generation, we realized that participants were more open to provide their perspectives first. Thus, we asked for their opinions and then requested them to reflect on results from the previous study.
- Data collection instruments and technologies [Description of instruments (e.g. interview guides, questionnaires)
We used a semi-structured interview guide. Details are described on Page 5 Line 6-11.
- and devices (e.g. audio recorders) used for data collection;
The following statement was added on Page 4 line 26:
The focus groups were audio recorded and transcribed; recordings and transcripts were stored on a secure server.
- Please clarify the in findings and evidence (e.g. quotes, field notes, text excerpts, photographs) to substantiate analytic findings using a figure or table. The authors add the orange area in the Figure 1. However, it does NOT link to the summarized data of interviews (the participants are not satisfied with IFHP).
Our data study objectives were two-fold. We first investigated the oral health experiences of humanitarian migrants, the findings of which are presented as a narrative in the results section. Second, we studied how they sought dental care in Canada, as migrants covered by IFHP. This is presented in Figure 1. We have not made an attempt to link the experiences with the pathways because of the focused group analysis methodology that was followed. The lack of good IFHP may have affected how they sought care, but that wasn’t what we were asking of them specifically (see further explanation in #11).
- The readers can’t find the data; how many participants are not satisfied, how many participants received partial care and how many participants do self-medication. Please add more details in the result section.
- Why don’t the authors add the data of previous study and analyze them in a cross-sectional study?
We believe pooling the data in a cross sectional study would not be possible because data was collected at two different points in time, several years apart, in two different ways (interviews vs focus groups).
Reviewer 3 Report
I acknowledge the authors for scientifically representing an important but bit unexplored topic. Migration is an important issue nowadays where forcefully migrated refugees suffer a lot from different health problems.
The introduction is well written, however, the authors should also highlight the oral health status of other refugee studies around the world, like: https://www.aihbonline.com/article.asp?issn=2321-8568;year=2021;volume=11;issue=1;spage=135;epage=136;aulast=Isha.
Please add a section on the strengths and limitations of the current study.
Author Response
We thank Reviewer 3 for reviewing our manuscript.
I acknowledge the authors for scientifically representing an important but bit unexplored topic. Migration is an important issue nowadays where forcefully migrated refugees suffer a lot from different health problems.
- The introduction is well written, however, the authors should also highlight the oral health status of other refugee studies around the world, like: https://www.aihbonline.com/article.asp?issn=2321-8568;year=2021;volume=11;issue=1;spage=135;epage=136;aulast=Isha.
We are pleased that you found the introduction well written. We’ve referenced the scoping review by Keboa et al., which is also referenced by the commentary on the Rohinga refugees. This scoping review provides an overview of the oral health of humanitarian migrants, globally. The below statement has been rephrased to reflect your recommendation (Page 2 Line 10):
Globally, the burden of oral diseases is high in humanitarian migrants, which is followed by high need for dental care [1].
- Please add a section on the strengths and limitations of the current study.
The section on strengths and limitations has been revised on Page 14 line 4:
As with any study, our study has important strengths and limitations. To begin, our work builds on the findings of a previous study, providing further insight on the dental care experiences of humanitarian migrants in Canada. To the best of our knowledge, this is the first study to report on the oral health of humanitarian migrants from Ontario. While we were not able to include participants from all countries of origin of humanitarian migrants arriving in Canada, we interviewed humanitarian migrants from the four main global regions of origin, and our focus groups were facilitated in the mother tongue of participants. The diversity in our study population has provided more detailed information to understand complex issues. With respect to the qualitative study design, our findings were not meant to be generalised to all migrants, but rather to describe the experience of the humanitarian immigrants recruited in the study. The use of focus group discussions was chosen over interviews as focus groups have the advantage of allowing participants to jointly brainstorm and debate ideas, opinions, and to build recommendations together.
Round 2
Reviewer 2 Report
The paper was overall improved. However, there are some issues. The paper needs to be revised.
- Please add the ethical approval numbers (L124).
- Please add the comments about limitation of the qualitative study compared to quantitative studies.
Author Response
Response to reviewer comments:
Reviewer 2
- Please add the ethical approval numbers (L124).
This statement was added to the ethics approval section:
"The study was approved by the University of Toronto’s Research Ethics Board (RIS Protocol No. 36911)."
- Please add the comments about limitation of the qualitative study compared to quantitative studies.
This statement was added to the discussion section:
"We acknowledge the limitation of qualitative study designs in that it is not possible to generalise the findings to the study population, however, our findings were not meant to be generalised to all migrants, but rather to describe the experience of humanitarian immigrants."